# Pathogenicity and Genomic Characterization of a Novel Genospecies, *Bacillus shihchuchen*, of the *Bacillus cereus* Group Isolated from Chinese Softshell Turtle (*Pelodiscus sinensis*)

**DOI:** 10.3390/ijms24119636

**Published:** 2023-06-01

**Authors:** Li-Wu Cheng, Omkar Vijay Byadgi, Chin-En Tsai, Pei-Chi Wang, Shih-Chu Chen

**Affiliations:** 1Department of Veterinary Medicine, College of Veterinary Medicine, National Pingtung University of Science and Technology, Pingtung 91201, Taiwan; ppm8888200@gmail.com (L.-W.C.); chinen@mail.npust.edu.tw (C.-E.T.); 2Southern Taiwan Fish Diseases Research Centre, College of Veterinary Medicine, National Pingtung University of Science and Technology, Pingtung 91201, Taiwan; 3International Degree Program of Ornamental Fish Technology and Aquatic Animal Health, International College, National Pingtung University of Science and Technology, Pingtung 91201, Taiwan; omkarbyadgi@mail.npust.edu.tw; 4Research Centre for Fish Vaccine and Diseases, College of Veterinary Medicine, National Pingtung University of Science and Technology, Pingtung 91201, Taiwan; 5Research Centre for Animal Biologics, National Pingtung University of Science and Technology, Pingtung 91201, Taiwan

**Keywords:** *Bacillus cereus* group, *Bacillus shihchuchen*, *Pelodiscus sinensis*

## Abstract

The Chinese softshell turtle (CST; *Pelodiscus sinensis*) is a freshwater aquaculture species of substantial economic importance that is commercially farmed across Asia, particularly in Taiwan. Although diseases caused by the *Bacillus cereus* group (Bcg) pose a major threat to commercial CST farming systems, information regarding its pathogenicity and genome remains limited. Here, we investigated the pathogenicity of Bcg strains isolated in a previous study and performed whole-genome sequencing. Pathogenicity analysis indicated that QF108-045 isolated from CSTs caused the highest mortality rate, and whole-genome sequencing revealed that it was an independent group distinct from other known Bcg genospecies. The average nucleotide identity compared to other known Bcg genospecies was below 95%, suggesting that QF108-045 belongs to a new genospecies, which we named *Bacillus shihchuchen*. Furthermore, genes annotation revealed the presence of anthrax toxins, such as edema factor and protective antigen, in QF108-045. Therefore, the biovar anthracis was assigned, and the full name of QF108-045 was *Bacillus shihchuchen* biovar anthracis. In addition to possessing multiple drug-resistant genes, QF108-045 demonstrated resistance to various types of antibiotics, including penicillins (amoxicillin and ampicillin), cephalosporins (ceftifour, cephalexin, and cephazolin), and polypeptides, such as vancomycin.

## 1. Introduction

*Pelodiscus sinensis*, also known as the Chinese softshell turtle (CST), is an important freshwater aquaculture species with high nutritional value [1,2,3]. CST farming provides substantial economic benefits to the aquaculture industry in Taiwan. Bacillosis caused by *Bacillus cereus* group (Bcg) strains poses a serious threat to commercial CST farming, owing to the high mortality rate [3]. In a previous study, we reported that infected turtles exhibited clinical signs such as epistaxis, paralysis, and petechial hemorrhages on the skin. Gross lesion findings revealed severe oedema in the body cavity, and a histological examination revealed sepsis, hemolysis, and heterophils infiltration [3]. 

Bcg, also known as *Bacillus cereus* sensu lato, is a group of bacteria that are Gram-positive, facultative anaerobic, spore-forming, and motile rods [4]. These bacteria are present in various environments and exhibit a phylogenetic similarity while demonstrating significant ecological and phenotypic variations [5,6]. Notable members include *Bacillus anthracis*, the pathogen that causes anthrax disease in humans [7,8]; *Bacillus cereus* sensu stricto, which is widely thought to be a foodborne pathogen, but has also been linked to anthrax-like symptoms [9,10,11]; and *Bacillus thuringiensis*, a well-known biopesticide [11,12]. Nevertheless, the above-mentioned phenotypic characteristics used in Bcg taxonomy may differ among species [13,14]. Additionally, plasmid-mediated genomic determinants are responsible for certain phenotypes, such as anthrax toxin/capsular protein synthesis [15,16,17], bioinsecticidal toxins (crystal proteins) [18,19], and emetic toxin synthetase proteins (cereulide) [20,21]. These features can be lost, acquired, or diverged within a species, or may be present in numerous species [22,23,24]. 

Modern species delineation processes utilize high-throughput average nucleotide identity (ANI) [25,26] and digital DNA-DNA hybridization (dDDH)-based [27,28] methodologies. When two genomes have an ANI and dDDH value greater than a pre-set threshold, they are considered as belonging to the same genomospecies (usually 95% ANI and 70% dDDH) [29,30,31,32]. Therefore, an accurate ontological framework for Bcg was built that adheres to widely accepted genomic and taxonomic definitions of bacterial genomospecies [33,34,35] while remaining insightful, straightforward, and comprehensible to those in public health by adding the biovar name (phenotype) [34]. 

In our previous study, we observed the presence of the hemolysin BL complex, enterotoxin FM, and non-hemolytic enterotoxin complex in Bcgs isolated from infected softshell turtle [3]. Additionally, certain Bcgs were found to carry a plasmid known as pXO1. This plasmid contains genes that encode important virulence factors, including protective antigen, edema factor, and lethal factor, which play a crucial role in the pathogenicity of anthrax in humans [15,36,37]. However, there is no clear description of plasmid in Bcgs isolated in CST; therefore, the identification of plasmids will be emphasized in this study.

Despite the substantial losses to CST farming caused by this disease in Taiwan, its pathogenicity, genomic sequence, and antibiotic susceptibility have yet to be analyzed. Therefore, we performed pathogenic studies of Bcg strains recently isolated from CSTs in Taiwan. To better understand this pathogen, we sequenced the genome using the Oxford nanopore MinIon platform, which has been used to identify aquaculture pathogens, such as *Edwardsiella anguillarum* and *Edwardsiella piscicida* [38], and has proven to be fast and reliable. Breakthrough results regarding the whole genome of Bcg isolated from CST can enable epidemiological research, the development of vaccines, and the identification of antibiotic-resistant genes. Similarly, antibiotic susceptibility tests can shed light on the efficacy of antibiotic treatment in the farming industry.

## 2. Results 

### 2.1. Pathogenicity of Bcg Strains

Based on a previous study, 57 Bcg strains isolated from diseased turtles were divided into 4 groups based on their genotypic characteristics: N1, S1, and E1; N2, S2, and E2; N3, S3, and E3; and N4, S4, and E4. One representative strain from each group (N1, S1, and E1-QF108-010), (N2, S2, and E2-QF108-011), (N3, S3, and E3-QF108-043), and (N4, S4, and E4-QF108-045) was used to confirm pathogenicity [3]. Clinical signs of CSTs from five groups (N1, S1, and E1-QF108-010), (N2, S2, and E2-QF108-011), (N3, S3, and E3-QF108-043), (N4, S4, and E4-QF108-045), and PBS (control) were recorded daily after the challenge. Limb weakness, epistaxis, local extensive hemorrhages on the skin, and eyelid oedema were the most common clinical signs in QF108-045-challenged turtles. Furthermore, some turtles also exhibited neck opisthotonus. Although the (N1, S1, and E1-QF108-010)-, (N2, S2, and E2-QF108-011)-, and (N3, S3, and E3-QF108-043)-challenged CSTs showed a loss of appetite, no other clinical signs were observed. No clinical signs were observed in the PBS-treated group. Serosanguineous fluid accumulation was observed in the gross lesions of (N4, S4, and E4-QF108-045)-challenged CSTs, along with kidney hemorrhage, splenomegaly, and intestinal serosa oedema, as shown in Appendix A. The mortality rate of the challenged CSTs is shown in Figure 1. QF108-045 caused the highest mortality in the CST until day 6, with a mortality rate of 90%. The QF108-010 and QF108-043 strains caused only 10% mortality, whereas the QF108-011 and PBS groups showed no mortality (Figure 1). Next, we use the most virulent strain, QF108-045, to evaluate its LD_50_ dose in CST. The LD_50_ dose of QF108-045 was 1.37 × 10^5^ CFU (colony-forming units)/turtle.

### 2.2. QF108-045 and Bcg Reference Strains Genomic Information

The total number of base pairs (bp) and GC content of the QF108-045 genome and plasmids were 5,247,466 bp and 35.55%, and 229,197 bp and 31.93%, respectively. The annotated genome yielded 6,950 CDSs with 105 tRNA and 42 rRNA in QF108-045 genomes, and 395 CDSs with no tRNA or rRNA for plasmids. Figure 2 shows circular genome maps representing the full genome assembly. The genome size, rRNA, tRNA, CDS, and plasmid of QF108-045’s similar strains and Bcg reference strains are listed in Table 1 and Appendix A (for detailed information). Bcg reference strains included *Bacillus mosaicus* (reference genome: *Bacillus anthracis* Ames (now *B. mosaicus* subsp. *anthracis* strain Ames)), *Bacillus cereus* sensu stricto (reference genome: *B. cereus* ATCC 14579 (now *B. cereus* sensu stricto strain ATCC 14579)), *Bacillus luti* (reference genome: *B. luti* TD41), *Bacillus toyonensis* (reference genome: *B. toyonensis* strain BCT-7112), *Bacillus pseudomycoides* (reference genome: *B. pseudomycoides* strain DSM 12442), *Bacillus mycoides* (reference genome: *Bacillus weihenstephanensis* (now *B. mycoides* strain WSBC 10204)), *Bacillus cytotoxicus* (reference genome: *B. cytotoxicus* strain NVH 391-98), and *Bacillus paramycoides* (reference genome: *B. paramycoides* strain NH24A2).

pBS01’s similar plasmids are listed in Table 2. It is apparent that pBS01 shows the greatest similarity to the plasmid found in *Bacillus* sp. SYJ. This strain was isolated from infected CST in China. Furthermore, it is noteworthy that pBS01 also exhibits significant similarity to plasmids found in other strains responsible for causing food poisoning and anthrax in humans, such as the pXO1 plasmid, which has been isolated from various *B. anthracis* strains.

### 2.3. Ribosomal Multilocus Sequence Typing (rMLST) and Multilocus Sequence Typing (MLST) for QF108-045

After typing QF108-045 for ribosome protein subunits (rps) genes, we recorded 45 of the 65 rps genes. Among the 45 rps genes analyzed, a significant number of them exhibited close matches with various Bcg strains, including *B. cereus*, *B. thuringiensis*, *B. anthracis*, and *B. tropicus* (Appendix A). The MLST results showed that two loci in the QF108-045 genome had two novel alleles, which were Glp and Pta (Appendix A); therefore, the sequence type (ST) it matched may represent the nearest one. Nevertheless, considering all housekeeping genes’ alleles, the QF 108-045 strain was still found to be most closely related to *B. cereus* ST 234 (Table 3).

### 2.4. Genome Annotation Using RAST

To further comprehend the QF108-045-annotated genes, the subsystem technology of the RAST server was used, which is associated with biological activities and metabolic pathways. The QF108-045 genome contained 577 subsystems and 7551 coding sequences (Appendix A). The most distinctive subsystems were amino acids and derivatives (459); carbohydrates (310); cofactors, vitamins, prosthetic groups, and pigments (250 genes); protein metabolism (186 genes); nucleosides and nucleotides (157 genes); DNA metabolism (113 genes); and respiration (83 genes) (Figure 3).

### 2.5. Whole-Genome ANI, dDDH Comparison, and Phylogenetic Analysis

Based on Appendix A, we conducted phylogenetic, dDDH, and ANI analyses for a comparison of QF108-045 with other known Bcg genospecies. The phylogenetic tree showed that the QF108-045 genome belonged to an independent group distinct from the other Bcg genospecies. In addition, the outermost bar chart in the phylogenetic tree listed the Bcg strains’ GC ratio, which was approximately around 35% (Figure 4). The ANI and dDDH results are listed in Figure 5 and Figure 6, respectively. Additionally, QF108-045, *Bacillus* sp. SYJ, *B. thuringiensis* serovar chanpaisis BGSC 4BH1, *B. thuringiensis* 261-1, and *B. cereus* AFS012518 had ANI and dDDH values higher than 95% and 70%, respectively, compared to other known genospecies, indicating that they belong to the same genospecies. Furthermore, when comparing their ANIs and dDDHs with other known genospecies, the results all showed values lower than 95% and 70%, respectively. These results suggest that they are novel and independent genospecies; therefore, we proposed a new genospecies, *Bacillus shihchuchen*.

### 2.6. Genomic Islands in QF108-045

We discovered several genomic islands (GIs) after annotating the whole genome of QF108-045 (Figure 7). Seventeen GIs comprising two hundred and thirty genes were recorded in chromosomal DNA. Among the 17, the 7th GI, starting from 2,971,997 to 2,994,351 with a size of 29,553 bp, was found to be the largest, and encoded the TetR family regulatory protein of the multidrug resistance (MDR) cluster, genes involved in heme utilization or adhesion, membrane components of the MDR system, CAAX amino terminal protease family protein, and ribonucleotide reductase transcriptional regulator. The second-largest GI was the 6th out of the 17 GI, starting from 2,816,153 to 2,835,404 with a size of 36,082 bp, which mainly encoded the hydrolase, HAD superfamily, nutrient germinant receptor hydrophilic subunit C, succinyl-CoA ligase, and phage tail fibre proteins (Appendix A).

Five GIs comprising two hundred and forty genes were detected in the plasmid DNA. Among the 5, the 1st GI starting from 5,281,216 to 5,359,300 with a size of 78,084 bp was found to be the largest, which encoded the S-layer protein, reticulocyte binding protein, calmodulin sensitive adenylate cyclase (edema factor, EF), N-Acetylneuraminate cytidylyltransferase, threonine dehydratase, and L-serine dehydratase. The second-largest GI was the 5th, starting from 5,366,176 to 5,390,220 with a size of 24,044 bp, which encoded the hemolysins and related proteins containing CBS domains, flagellar P-ring protein (FlgI), mobile element protein, and magnesium and cobalt efflux protein (CorC) (Appendix A).

### 2.7. Distribution of Likely Virulence Genes in QF108-045 Genomes

Based on the BLAST results of the protein sequences in the VFDB, we identified several virulence genes located on the QF 108-045 chromosome and plasmid (Appendix A). For example, hemolysin III, hemolytic enterotoxin HBL, non-hemolytic enterotoxin, and O-antigen exist in the chromosome, whereas anthrax toxin (EF encoded by cya gene and protective antigen, PA, encoded by pagA gene), exopolysaccharide (BPS), and hyaluronic acid capsule-related genes exist in the plasmid. Using the information from the VFDB and RAST annotations, we identified various genes for evaluation of their localization, secretion, solubility, and antigenicity. Our screening process involved genes associated with invasion, immune evasion, motility, and outer membrane proteins, and proteins with antigenicity scores greater than 0.5 were reported. Several proteins from QF108-045 have been identified as promising subunit proteins for vaccine development. Appendix A includes information on these proteins, including their annotated genomes, antigenicity scores, and solubilities. The open reading frames of these proteins are listed in Appendix A. Because QF108-045 contains the cya and pagA virulence genes, the biovar anthracis was assigned. Therefore, the bacterium was designated as *B. shihchuchen* biovar anthracis.

### 2.8. Comparative Proteomics for Identification of Virulence Genes from QF108-045

*B. shihchuchen* biovar anthracis QF108-045 isolated from CST is closely related to *B. thuringiensis* serovar chanpaisis strain BGSC 4BH1 and *B. thuringiensis* strain 261-1, both of which were isolated from rice paddies and soil and are non-pathogenic. A comparison of the proteomic differences between non-pathogenic and pathogenic (QF108-045) bacteria revealed that certain unique proteins, such as N-acetylneuraminate synthase, S-layer homology domain protein, reticulocyte-binding protein, cya, and the transcriptional activator AtxA, appeared only in QF108-045. Furthermore, these protein-encoding genes were identified in plasmids. Additionally, some proteins, such as collagen adhesion, fibronectin type III domain protein, spore coat protein CotX, and CotY, only appeared in the chromosome of QF108-045. All the proteins that only existed in QF108-045 are shown in Figure 8.

### 2.9. Antibiotic-Resistant Genes

Antibiotic-resistant genes were identified in the present study. Several antibiotic-resistant genes were detected, including *B. cereus* beta-lactamase I, class A (BcI), *B. cereus* beta-lactamase II, class B (BcII family), fosfomycin-resistant enzyme (FosB), small multidrug resistance (SMR) efflux pump (qacJ), Streptomyces rimosus oxytetracycline resistance ribosomal protection protein otr (A), the serine racemases vanW, vanT, and vanY, and ATP-binding cassette, antibiotic efflux pump (RanA) (Figure 9).

### 2.10. Antibiotic Susceptibility Test

The antibiotic susceptibility of QF108-045 to 25 antimicrobial agents was screened to evaluate the sensitivity of antimicrobial drugs against QF108-045. Our data showed that QF108-045 was resistant to three penicillins (penicillin, amoxicillin, and ampicillin), three aminoglycosides (gentamycin, amikacin, and tobramycin), three cephalosporins (ceftifour, cephalexin, and cephazolin), tetracycline, phenicol (thiamphenicol), and a glycopeptide (vancomycin). However, it was susceptible to penicillin (piperacillin), tetracycline (doxytetracycline), phenicol (chloramphenicol), two quinolones (norfloxacin and ofloxacin), and macrolide (erythromycin) (Table 4).

## 3. Discussion

The purpose of this study was to use whole-genome sequencing to determine the pathogenicity of Bcg strains isolated from diseased CSTs and determine their genospecies, virulence genes, and antibiotic-resistant genes. Additionally, we performed an antibiotic susceptibility test to examine the susceptibility of the isolated strain to various antibiotic types and generations, and the results provide insights into the clinical treatment of this pathogen.

In a previous study, Pulsed-field Gel Electrophoresis (PFGE) and Enterobacterial Repetitive Intergenic Consensus (ERIC) PCR were used to confirm the genotypes of 57 clinical Bcg isolates from diseased CSTs [3]. The isolates were divided into four genogroups: N1, S1, and E1; N2, S2, and E2; N3, S3, and E3; and N4, S4, and E4. N denotes the PFGE result of *Not*I restriction enzyme-cut chromosome, S denotes *Sma*I restriction enzyme-cut chromosome, E denotes ERIC PCR, and 1, 2, 3, 4 denote different patterns. We found that 54 of the 57 strains belonged to the N4, S4, and E4 genotypes, and only one strain was found to belong to the genotypes N1, S1, and E1; N2, S2, and E2; and N3, S3, and E3 [3]. The pathogenicity study showed that the N4, S4, and E4 genotype (QF108-045) caused the highest mortality rate (90%) in CSTs (Figure 1). The etiology of Bcg infection in CSTs can be established based on Koch’s postulates [39].

Several approaches have been used to define bacterial species (e.g., 16S rRNA gene sequencing and phenotypic characterization) [9,40]. Nevertheless, modern species delineation approaches utilize high-throughput ANI and dDDH-based methodologies [25,27,28], meaning that two genomes belong to different genera if they share an ANI value lower than a predefined threshold (usually 95% ANI and 70% dDDH) [29]. In addition, the taxonomic approach commonly incorporates the comparison of whole genome nucleotides’ GC ratio. However, in the case of Bcg strains, the GC ratio was approximately around 35% (Figure 6), which renders the use of the GC ratio for the taxonomic classification of Bcg impossible.

A previous study has shown that Bcg genomes can be grouped Into a taxonomic framework that is flexible enough to consider phenotypes, making it easier for the public to understand [34]. These genospecies include: (I) *B. pseudomycoides* (reference genome: *B. pseudomycoides* strain DSM 12442); (II) *B. paramycoides* (reference genome: *B. pseudomycoides* strain NH24A2); (III) *B. mosaicus* (reference genome: *B. anthracis* Ames (now *B. mosaicus* subsp. *anthracis* strain Ames), *B. mobilis* (now *B. mosaicus* strain 0711P9-1), *B. pacificus* (now *B. mosaicus* strain EB422), *B. paranthracis* (now *B. mosaicus* strain MN5), and *Bacillus wiedmannii* (now *B. mosaicus* strain FSL W8-0169)); (IV) *B. cereus* sensu stricto (reference genome: *B. cereus* ATCC 14579 (now *B. cereus* sensu stricto strain ATCC 14579)); (V) *B. toyonensis* (reference genome: *B. toyonensis* strain BCT-7112); (VI) *B. mycoides* (reference genome: *B. mycoides* strain DSM 2048 and *B. weihenstephanensis* (now *B. mycoides* strain WSBC 10204)); (VII) *B. cytotoxicus* (reference genome: *B. cytotoxicus* strain NVH 391-98); and (VIII) *B. luti* (reference genome: *B. luti* TD41) [33,34,35]. At an ANI threshold of 95%, all genomes assigned to the aforementioned species based on their reference genomes or respective type strains belonged to that genospecies [34]. Interestingly, phylogenetic analysis revealed *B. shihchuchen* that was independent of all other known Bcg genospecies. Additionally, although *B. mosaicus* is the most closely related genospecies, all *B. mosaicus* strains exhibited ANI values of less than 95% in comparison to the newly identified genospecies, *B. shihchuchen*. This provides evidence for classifying *B. shihchuchen* as a novel genospecies (reference genome: *B. shihchuchen* strain QF108-045).

Based on the RAST annotating, bacterial GI, and virulence factor detection, the chromosome contained the TetR family regulatory protein of the MDR cluster, genes involved in heme utilisation or adhesion, and other virulence genes, including hemolysin III, hemolytic enterotoxin HBL, non-hemolytic enterotoxin, and O-antigen. In the plasmid, we identified virulence genes, including anthrax toxin (EF and PA) and exopolysaccharides. These results indicate the importance of both chromosomes and plasmids in contributing to pathogenesis (Figure 7, Appendix A).

The anthrax tripartite toxin is composed of EF, PA, and lethal factor [41]. The QF108-045 plasmid was found to contain *cya* and *pagA* genes. Therefore, the biovar anthracis was assigned, and the full name of QF108-045 was *B. shihchuchen* biovar anthracis. By converting ATP to cAMP, EF operates as a calmodulin-dependent adenylyl cyclase, substantially increasing intracellular cAMP levels and causing cardiovascular dysfunction and tissue damage [42,43]. PA can transport EF inside the cell through its binding affinity for the host cell’s TEM8/ATR receptors. Subsequently, anthrax toxin-receptor complexes are internalized and transported to early endosomes through clathrin-coated pits [36]. These factors may contribute to the clinical signs and gross lesions of QF108-045-challenged CSTs, such as eyelid oedema and serosanguineous fluid accumulation in the body cavity (Appendix A). 

Upon examining Table 2, it becomes evident that the *B. shihchuchen* biovar anthracis pBS01 exhibits the closest relationship to *Bacillus* sp. SYJ’s plasmid [44]. This strain was isolated from Chinese soft-shelled turtles in China and is associated with septicaemia infection. It is worth noting that pBS01 also exhibited close relatedness to other strains causing food poisoning and anthrax in humans. Therefore, further experimental studies are required to investigate the zoonotic potential of *B. shihchuchen* biovar anthracis QF108-045.

Prophylactic vaccines are the most cost-effective treatment approach for the effective treatment and prevention of bacillosis. In bacillus-related vaccine development, a well-studied vaccine against *B. anthrax* is used in human medicine [45,46]. PA is considered an important antigen in *B. anthrax* development. Immunity against an anthrax infection requires a sufficient humoral response to PA [47,48]. Anti-PA antibodies have been demonstrated to block the early stages of spore infection and neutralize anthrax toxin activity [49]. PA, which is encoded by *pagA*, was also observed in QF108-045. According to the ANTIGENpro prediction, PA also has a very high point of antigenicity (0.9), indicating that PA might be a promising antigen for future vaccine development against QF108-045. In addition to PA, other virulence genes whose products share high antigenicity are listed in Appendix A, such as hemolytic enterotoxin, non-hemolytic enterotoxin, and reticulocyte binding protein. The antigen candidate screening conducted in this study will aid in future vaccine development. 

We observed that most strains of *B. shihchuchen* genospecies were non-pathogenic, and were isolated from soil or agricultural products, such as weeds and rice paddies; nevertheless, some were pathogens of CSTs. Among *B. shihchuchen* genospecies, *B. shihchuchen* biovar anthracis QF108-045 was most closely related to *Bacillus thuringiensis* serovar chanpaisis BGSC 4BH1 (now *B. shihchuchen* serovar chanpaisis BGSC 4BH1) and *Bacillus thuringiensis* 261-1 (now *B. shihchuchen* 261-1), with ANIs of 99.47 and 99.45, respectively. One was a pathogenic bacterium isolated from CSTs, and the other was a non-pathogenic bacterium isolated from rice paddies and weeds. N-acetylneuraminate synthase, N-acetylneuraminate cytidylyltransferase, EF, and PA were found only in QF108-045 (Figure 8). The synthesis of N-acetylneuraminic acid (Neu5Ac or sialic acid) is catalyzed by N-acylneuraminate cytidylyltransferase, which converts Neu5Ac into diphosphate and CMP-N-acylneuraminate [50]. Activated CMP-N-acylneuraminate is then incorporated into capsular polysaccharides [51]. In *Streptococcus pneumoniae*, the polysaccharide capsule is the primary surface structure of the organism and plays an important role in virulence, mostly by interfering with the host opsonophagocytic clearance systems [52]. Furthermore, bacterial cell-surface sialic acids are thought to mimic host sialoglycoconjugates, allowing microorganisms to avoid detection by the host’s innate immune system [53,54]. The capsular polysaccharides of QF108-045 require further investigation in future pathogenic studies. 

Antibiotic-resistant genes were located in the 7th GI starting from 2,971,997 to 2,994,351 on the chromosome (Figure 7). We observed BcI and BcII family genes, which render bacteria resistant to beta-lactam antibiotics, such as penicillins and cephalosporins [55]. Although QF108-045 was resistant to first-(penicillin) and second-generation penicillins (amoxicillin and ampicillin), it was susceptible to fourth-generation penicillins, such as piperacillin [56]. Other antibiotic-resistant genes included *Streptomyces rimosus* otr (A) and vanW, vanT, and vanY, which may confer resistance to tetracycline [57] and vancomycin [58]. Notably, qacJ is an SMR efflux pump that confers resistance to the quaternary ammonium compound [59]. Alkyldimethylbenzylammonium chloride (BKC), a quaternary ammonium compound, is a disinfectant typically used by CST farmers. Further research on the resistance of QF108-045 to BKC is required.

## 4. Materials and Methods

### 4.1. Animals

This study was performed using CSTs (200 ± 10 g). To confirm that the experimental turtles were bacillosis-free, bacterial isolation and polymerase chain reaction (PCR) examinations of the livers, spleens, and kidneys were performed on five sacrificed CSTs prior to importation [3]. The CSTs were kept in a freshwater outdoor facility at 28 ± 2 °C with adequate sunlight; 30% of the water was changed daily, and 3% body weight of commercial dry pellet was fed to the CSTs twice daily. Before performing the experiment, the CSTs were acclimatized for 1 week. 

### 4.2. Experimental Challenge and Median Lethal Dose (LD_50_)

One representative strain from each group (N1, S1, and E1-QF108-010), (N2, S2, and E2-QF108-011), (N3, S3, and E3-QF108-043), and (N4, S4, and E4-QF108-045) was used to confirm pathogenicity [3]. The strains were grown on brain–heart infusion agar (BHI, Becton Dickinson, Auvergne-Rhône-Alpes, France) at 25 °C for 24 h, and were subsequently suspended in sterile phosphate-buffered saline (PBS; Na_2_HPO_4_·12H_2_O, NaH_2_PO_4_·2H_2_O, NaCl) to achieve optical density 1 at an absorbance of 600 nm. The bacteria were counted using the serial dilution method [60], and 20 μL of 10-fold diluted samples were dropped on a BHI agar plate. After 24 h of incubation at 25 °C, the colony forming units per ml (CFU/mL) were determined. Fifty CSTs were randomly placed in five tanks and anesthetized by administering 100 mg/L of tricaine methane sulphonate (MS-222). Subsequently, 0.1 mL of bacterial suspension was intraperitoneally (IP) injected into each turtle (10^7^ CFU/turtle). The same treatments were administered to the control CST, except that they received 0.1 mL of sterile PBS. Behavioral and other clinical indicators were consistently observed in the CSTs, and the number of deaths was recorded over 14 days. Subsequently, bacteriological and gross examinations were performed on the deceased turtles. PCR was used to identify the isolated bacteria. The LD_50_ in the CST was evaluated by the IP injection of the following bacterial concentrations (10^3^, 10^4^, 10^5^, and 10^6^ CFU/turtle), plus one PBS group. Ten CSTs were assigned to each group, and each group was injected with 0.1 mL of bacterial suspension or 0.1 mL of sterile PBS. The LD_50_ was calculated using Beheren’s method [61]. 

### 4.3. Genomic DNA Extraction and DNA Quality Control

A Quick-DNA Bacterial Miniprep Kit (Zymo Research, Irvine, CA, USA) was used to extract genomic DNA from actively growing isolates. The quality of the extracted DNA was evaluated using 1% agarose gel (Bio-Helix) electrophoresis (100 V, 20 min), and images were acquired using a Gel Doc^TM^ XR^+^ Imager System (Bio-Rad Laboratories). A NanoPhotometer was used to measure absorbance at 260 and 280 nm (Implen, Westlake Village, CA, USA). For a good-quality DNA sample, A260/A280 and A260/A230 ratios of 1.8–2.0 and 2.0–2.2, respectively, are acceptable. Only samples that met the inclusion criteria were used in the next stage of the experiment.

### 4.4. DNA Library Preparation and Sequencing Using MinIon Nanopore MK1C

A DNA library was created using the Rapid Sequencing Kit SQK-RAD004 (Oxford Nanopore Technologies, Oxfordshire, UK). A fragmentation mix containing 400 ng of genomic DNA was used for random fragmentation. After incubation at 30 °C for 1 min and then at 80 °C for 1 min in a thermal cycler (Takara Bio, San Jose, CA, USA), the mixture was placed on ice and chilled at 4 °C. The fast adapter was incubated with the fragmented genomic DNA for 5 min at room temperature, and the prepared DNA library was stored on ice until being placed in a flow cell. Prior to each sequencing read, flow cell (FLO-MIN106D; Oxford Nanopore Technologies, UK) quality check (QC) was performed on a MinION Mk1C sequencer (Oxford Nanopore Technologies). After passing the QC, the flow cell was then primed with a priming kit (ONT EXP-FLP002; Oxford Nanopore Technologies). The flush buffer, flush tether, loading beads, and sequencing buffer were slowly thawed on ice for loading in the flow cell. The DNA library was then added to a flow cell placed on a MinION Mk1C sequencer for 12 h, according to the manufacturer’s protocol. The flow cell was cleaned using a Flow Cell Wash Kit (ONT EXP-WSH004; Oxford Nanopore Technologies, Oxfordshire, UK) after each run and maintained at 4 °C for future use.

### 4.5. Raw Data Pre-Processing and Genome Assembly

All fast5 files were base-called using Guppy (version 4.3.4, Oxford Nanopore Technologies) to create fastq files when adequate reads were obtained from the sequencing runs. The ONT Gussy base-calling tool was used to transform raw signals into a DNA sequence (version 4.2.3, Oxford Nanopore Technologies). Canu [62] and Flye [63] were used for de novo genome assembly. All sequences were uploaded to the National Center for Biotechnology Information (NCBI, https://www.ncbi.nlm.nih.gov, accessed on 20 April 2023) with the BioProject and BioSample accession number PRJNA948896 and SAMN33923097, respectively.

### 4.6. Functional Enrichment Analysis

Briefly, the Prokka pipeline V1.14.5 [64] was used to predict ribosomal RNA (rRNA), transfer messenger RNA (tmRNA), and transfer RNA (tRNA) genes and coding sequences (CDSs) in the genomes. Ribosomal Multilocus Sequence Typing (rMLST) was performed using public databases for molecular typing and microbial genome diversity (PubMLST; https://pubmlst.org/, accessed on 29 March 2023). The ST was identified using the Multilocus Sequence Typing (MLST) database (http://www.genomicepidemiology.org/, accessed on 29 March 2023). Antibiotic-resistant genes were identified using the Comprehensive Antibiotic Resistance Database (https://card.mcmaster.ca/, accessed on 29 March 2023). Whole-genome sequencing results were uploaded to the Rapid Annotations using the Subsystems Technology (RAST, https://rast.nmpdr.org/rast.cgi, accessed on 30 March 2023) [65] server to annotate CDSs for subsystem mapping. The RAST-annotated findings were visualized using a SEED viewer [65]. Subsequently, a circular genome map was constructed using the ‘circular viewer’ functionality built within the Pathosystems Resource Integration Center (PATRIC version 3.6.2) web server [66,67].

### 4.7. Whole-Genome Distance, ANI, and dDDH Comparsion

To identify QF108-045′s similar genomes and plasmids, we utilized Mash [68], a MinHash dimensionality-reduction technique. This method enabled us to condense large sequences or sets of sequences into compressed sketch representations [69]. The computation of Mash distance can be rapidly performed using the size-reduced sketches, while still yielding highly correlated results with alignment-based metrics, such as ANI [70]. The whole-genome ANI comparison was conducted using the average nucleotide identity calculator online server (http://enve-omics.ce.gatech.edu/ani/, accessed on 11 April 2023) created by Rodriguez-R et al. [71]. This server utilizes both best hits (one-way ANI) and reciprocal best hits (two-way ANI) approaches when comparing two sets of genomic data.

Digital DNA-DNA hybridization was analyzed using the genome-to-genome distance calculator (GDDC) [72] with the Genome Blast Distance Phylogeny (GBDP) approach [73]. The GBDP used a basic local alignment search tool to locally align two genomes [74], yielding a list of high-scoring segment pairs (HSPs), before these HSPs were translated into a single genome-to-genome distance value. The resulting distances were then converted into values similar to DNA-DNA hybridization (DDH) using a generalized linear model, producing dDDH values. The GBDP formula d4, the sum of all the identities found in HSP divided by the overall HSP length, was applied for the dDDH value [73].

### 4.8. Whole-Genome Phlogenetic Tree

A whole-genome phylogenetic tree was generated using the PATRIC phylogenetic tree building service, which utilizes nucleotide sequences and amino acids from a collection of Bacterial and Viral Bioinformatics Resource Center (BV-BRC) [75] (https://www.bv-brc.org/, accessed on 10 April 2023) global Protein Families (PGFams) [76]. Both the nucleotide and amino acid sequences were used for mapping each of the PGFam-selected genes. MUSCLE was used to align protein sequences [77] and the codon alignment function of BioPython was used to align nucleotide-coding gene sequences [78]. All nucleotides and proteins were concatenated into an alignment and formatted in PHYLIP before being converted into a partition file for RaxML [79]. RaxML support values were generated using 100 rounds of the “Rapid” bootstrapping option [80]. Finally, the tree was graphically visualized using Interactive Tree Of Life software (version 6.7.5) [81].

### 4.9. Genomic Islands in QF108-045

To identify genomic islands in QF108-045 genomes, we utilized IslandViewer (version 4), a software developed by Bertelli et al. [82]. The GBK files were obtained from the RAST annotation server and compared to *Bacillus* sp. SYJ as a reference. The genomic island prediction tools, such as IslandPick and SIGI-HMM, were used to identify genomic islands, and associated mobility genes were determined using IslandPath-DIMOB and Islander tools [83].

### 4.10. Comparative Proteomics

Comparative proteomics was conducted using the Comparative Systems Service of BV-BRC [75]. This tool uses the protein family PATtyFams [76], annotated by RASTtk [66]. Finally, we used the Protein Family Sorter tool in the BV-BRC to study the distribution of certain protein families across various genomes, and generated a heat map for comparative proteomics.

### 4.11. Identification of Virulence Genes

Genomic sequences were matched against the virulence factor database (VFDB) via VFanalyser [84]. PSORTb (version 3.0.3), a protein subcellular localization prediction tool, was used to predict the protein localization and secretion [85]. All virulence genes were identified by inputting their protein sequences into the SCRATCH protein predictor [86]. The localized genes identified in the outer membrane or periplasmic regions with an antigenicity score greater than 0.5 (as identified by ANTIGENpro) were evaluated for their secretion score (identified by SOLpro), as previously described [38].

### 4.12. Antibiotic Susceptibility Test

The antimicrobial susceptibility of the CST isolate was assessed using the Kirby–Bauer disk diffusion procedure in accordance with the Clinical and Laboratory Standards Institute’s performance standards for antimicrobial susceptibility testing (The Clinical and Laboratory Standards Institute [CLSI], 2010/2015) for *Staphylococcus aureus* and *Enterococcus* spp., as previously described [87,88]. Twenty-five antibiotics (Abtek Biologicals, Liverpool, UK and BD BBL™, San Jose, CA, USA) were tested, including four penicillins (penicillin, amoxicillin, ampicillin, and piperacillin), three cephalosporins (ceftifour, cephalexin, and cephazolin), five aminoglycosides (neomycin, gentamycin, streptomycin, amikacin, and tobramycin), one nitrofuran (nitrofurantoin), three quinolones (enrofloxacin, norfloxacin, and ofloxacin), three tetracyclines (tetracycline, oxytetracycline, and doxycycline), one glycopeptide (vancomycin), three phenicols (chloramphenicol, florfenicol, and thiamphenicol), and two macrolides (erythromycin and clindamycin). After incubating the bacteria on Mueller–Hinton agar (BD BBL™, San Jose, CA, USA) for 24 h at 37 °C, the inhibition zone diameters were measured.

## 5. Conclusions

The present study confirmed the pathogenicity of Bcg strains isolated from CSTs and established the etiology of Bcg infection in CSTs. The whole-genome sequencing of the most virulent strain, QF108-045, confirmed the presence of a novel genospecies within Bcg, which we named *B. shihchuchen*. Furthermore, gene annotation revealed anthrax toxins, such as EF and PA, in QF108-045. Finally, as QF108-045 is multidrug-resistant, particularly to penicillins, aminoglycosides, and cephalosporins, antibiotic susceptibility testing prior to the treatment of CSTs with bacillosis is important. 

## Figures and Tables

**Figure 1 ijms-24-09636-f001:**
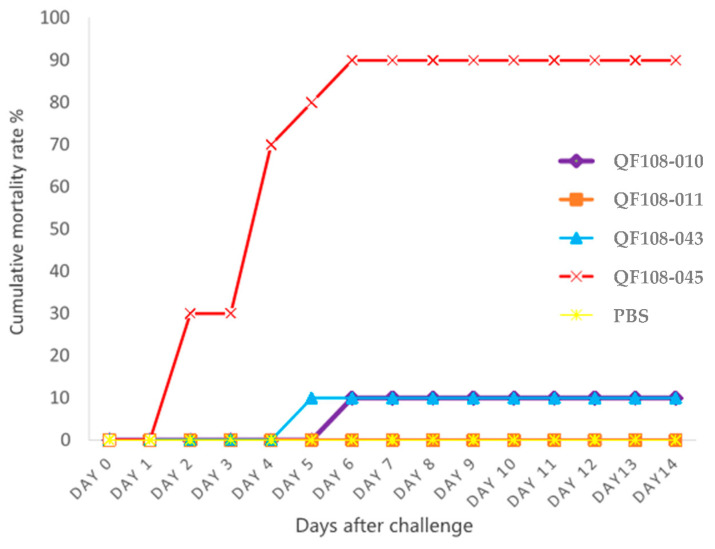
Cumulative mortality of Chinese softshell turtles challenged with different *Bacillus cereus* group strains, which included genotypic groups (N1, S1, and E1-QF108-010), (N2, S2, and E2-QF108-011), (N3, S3, and E3-QF108-043), and (N4, S4, and E4-QF108-045).

**Figure 2 ijms-24-09636-f002:**
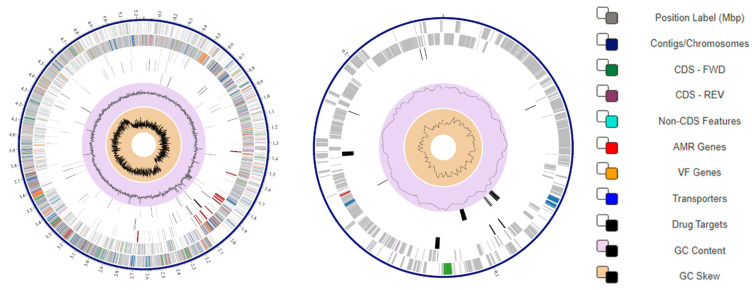
The graphical circular view of the QF 108-045 genome (**left**) and its plasmid (**right**). The tracks in the figure are displayed as concentric rings, from outermost to innermost tracks representing the 1-reference position in the genome, 2-position, and order of the assembled contigs, 3-CDS- forward strands, 4-CDS- reverse strand, 5-non-Coding features, 6-AMR (anti-microbial resistance) genes, 7-genes for virulence factors, 8-genes for transporters, 9-genes for drug targets, 10-GC content, and 11-GC skew, respectively.

**Figure 3 ijms-24-09636-f003:**
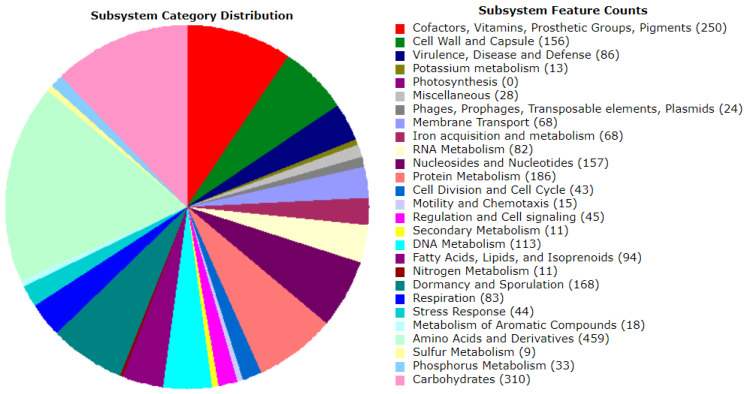
Summary of the subsystem categories referred to the genes predicted in QF108-045 genome. RAST (https://rast.nmpdr.org/rast.cgi, accessed on 29 March 2023) was used to annotate the whole genome sequence.

**Figure 4 ijms-24-09636-f004:**
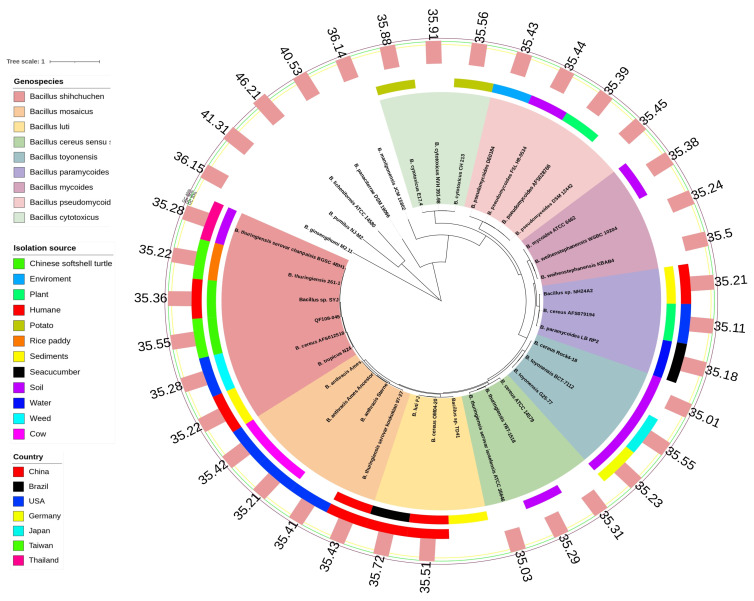
Phylogenetic analysis of QF 108-045 with other Bcg strains based on 290 genes. Different genospecies of Bcg are displayed in different colored range (**left upper legend**) within phylogenetic tree. Different color strips from inner to outer denote isolation source (**left middle legend**) and country (**left lower legend**) where strains are isolated. The outermost bar chart denotes the GC ratio (%) of different strains. The details of all the genomes used in this analysis and their NCBI BioProject accession numbers are listed in Appendix A. The statistical details of phylogenetic tree are listed in Appendix A.

**Figure 5 ijms-24-09636-f005:**
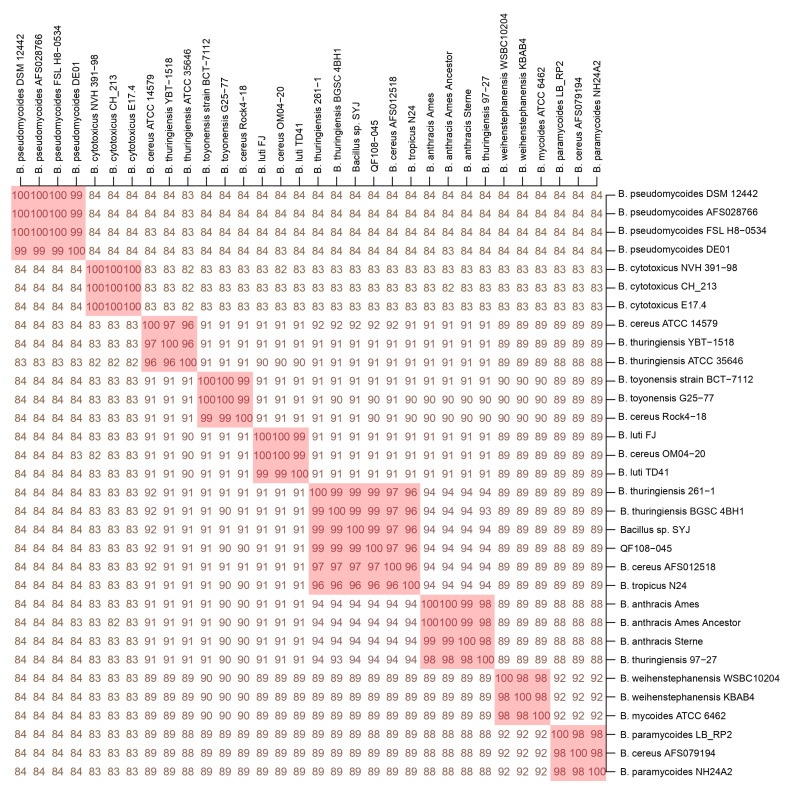
ANI comparison of *B. shihchuchen* biovar anthracis QF108-045 with other *Bacillus cereus* group genospecies. Note: Pink background represents the dDDH are above 70%. The genospecies (pink background) from upper left to lower right are *B. pseudomycoides*, *B. cytotoxicus*, *B. cereus* sensu stricto, *B. toyonensis*, *B. luti*, *B. shihchuchen, B. mosaicus*, *B. mycoides*, and *B. paramycoides*, respectively.

**Figure 6 ijms-24-09636-f006:**
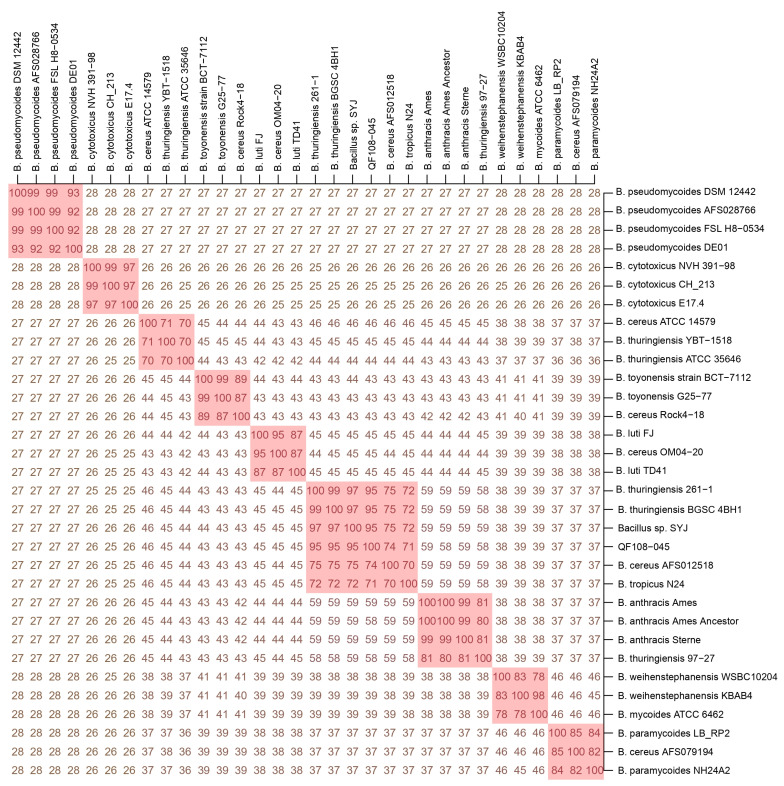
dDDH comparison of *B. shihchuchen* biovar anthracis QF108-045 with other *Bacillus cereus* group genospecies. Note: Pink background represents the dDDH are above 70%. The genospecies (pink background) from upper left to lower right are *B. pseudomycoides*, *B. cytotoxicus*, *B. cereus* sensu stricto, *B. toyonensis*, *B. luti*, *B. shihchuchen, B. mosaicus*, *B. mycoides*, and *B. paramycoides*, respectively.

**Figure 7 ijms-24-09636-f007:**
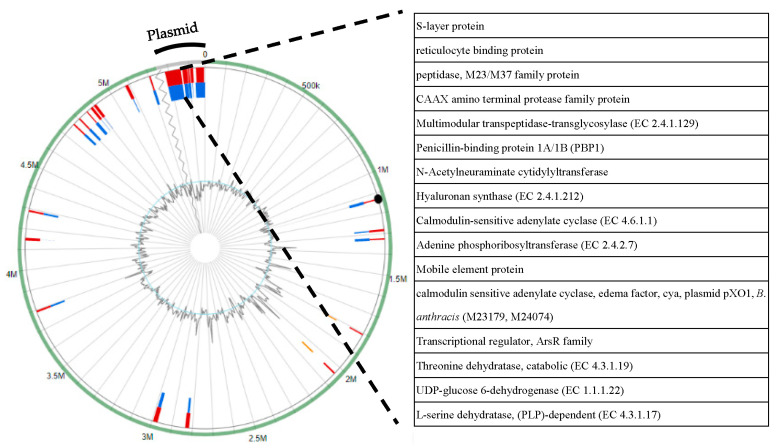
Predictions of genome island (GI) in QF108-045. The circle displays a chromosome and plasmid genome, with the outermost dark red and blue bars representing expected GI positions using IslandViewer 4 detection algorithms. Within the circle (starting from the outside toward the center), GIs predicted by both softwares IslandPath-DIMOB and SIGI-HMM are shown as red, and GIs predicted by software SIGI-HMM only are shown as blue. The genes of 1st GI in the plasmid are demonstrated in the right box.

**Figure 8 ijms-24-09636-f008:**
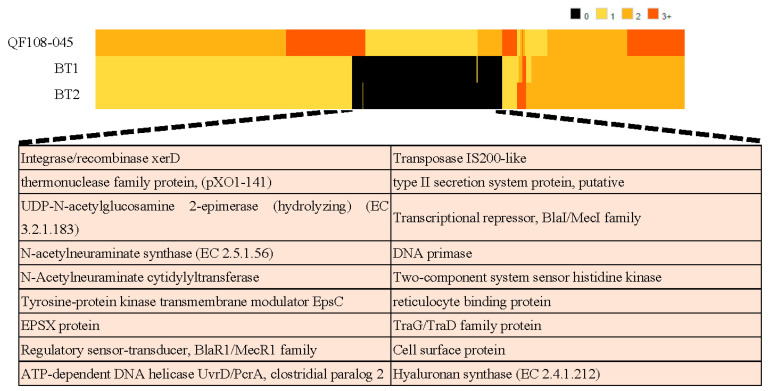
Comparative proteomics for identification of virulence genes from QF108-045. In the heat map, 0 (black) denotes the gene is absent in genome, 1 (light yellow) denotes the gene appeared one time, 2 (yellow) denotes the gene appeared two times, and 3 (dark yellow) denotes the gene appeared three times or more. In bottom rows, pink background denotes the protein-encoded genes existed in plasmid, grey background denotes the protein-encoded genes existed in chromosome, and green background denotes the protein-encoded genes existed in both plasmid and chromosome. BT1: *B. thuringiensis* serovar chanpaisis strain BGSC 4BH1, BT2: *B. thuringiensis* strain 261-1.

**Figure 9 ijms-24-09636-f009:**
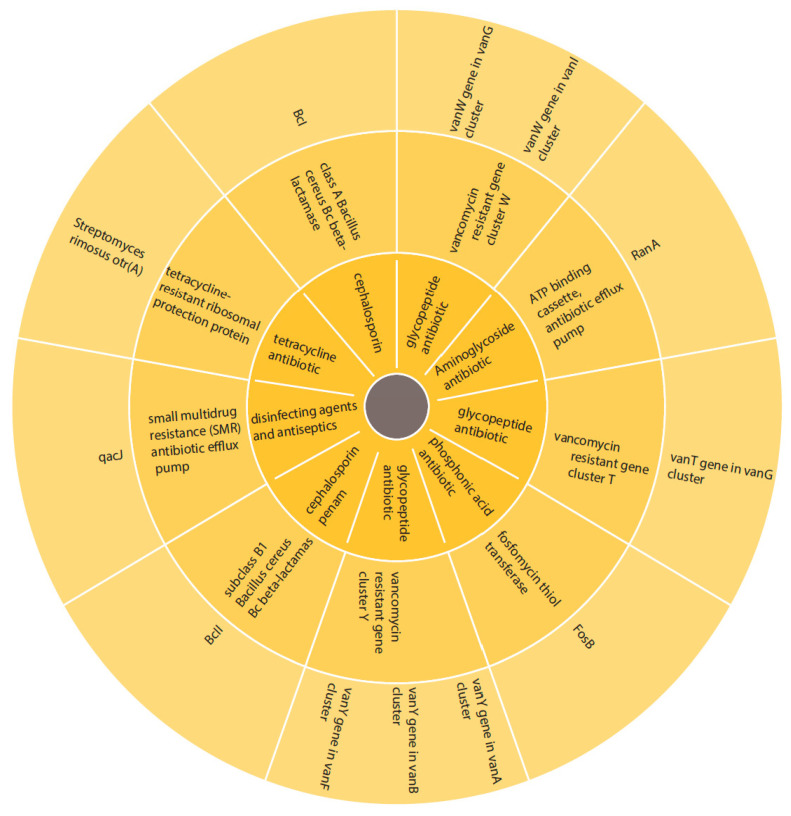
Pie chart that displays the antibiotic-resistant genes identified in QF108-045. From the outermost to innermost circle, the chart denotes the antibiotic-resistant genes, antibiotic-resistant genes’ family, and antibiotic family that the resistant genes targeted.

**Table 1 ijms-24-09636-t001:** Details of *Bacillus shihchuchen* biovar anthracis QF108-045 and other reference *Bacillus cereus* group genomic features.

Genome Name	*B. shihchuchen* Biovar Anthracis QF108-045 ^1^	*Bacillus* sp. SYJ ^1^	*B. thuringiensis* 261-1 ^1^	*B. thuringiensis* Serovar Chanpaisis BGSC 4BH1 ^1^	*B. cereus* AFS012518 ^1^	*B. anthracis* Sterne ^2^	*B. anthracis* ‘Ames Ancestor’ ^2^	*B. anthracis* Ames ^2^
Size	5,247,466	5,520,756	5,613,888	5,481,202	5,225,847	5,228,663	5,503,926	5,227,293
tRNA	105	106	73	90	75	95	95	95
rRNA	42	28	3	4	3	22	22	22
CDS	6950	5680	6224	5958	5649	5698	6010	5699
MLST	*Bacillus cereus*.234	*Bacillus cereus*.234	*Bacillus cereus*.234	*Bacillus cereus*.234	-	*Bacillus cereus.*1	*Bacillus cereus*.1	*Bacillus_cereus*.1
Plasmids	1	2	-	-	-	1	2	2
BioProject Accession	PRJNA948896	PRJNA523077	PRJNA387206	PRJNA349211	PRJNA400804	PRJNA10878	PRJNA10784	PRJNA309
**Genome Name**	***B. cereus* ATCC 14579 ^3^**	***Bacillus luti* TD41 ^4^**	***B. toyonensis* BCT-7112 ^5^**	***B. weihenstephanensis* WSBC 10204 ^6^**	** *B. mycoides* ** **ATCC 6462 ^6^**	***B. pseudomycoides* DSM 12442 ^7^**	***B. cytotoxicus* NVH 391-98 ^8^**	***Bacillus paramycoides* NH24A2 ^9^**
Size	5,427,083	5,094,423	5,025,419	5,608,349	5,637,053	5,782,514	4,094,159	5,094,423
tRNA	108	17	97	77	107	86	106	18
rRNA	26	2	36	7	36	2	26	2
CDS	5666	5400	5186	5838	5837	5367	4213	5849
MLST	*Bacillus cereus*.921	*Bacillus cereus.*764	-	-	*Bacillus cereus*.116	-	-	*Bacillus_cereus.*780
Plasmids	1	-	2	0	3	0	1	-
BioProject Accession	PRJNA384	PRJNA325892	PRJNA225857	PRJNA258373	PRJNA238211	PRJNA29707	PRJNA13624	PRJNA326288

Note: numeral superscripts denote the following genospecies: ^1^ *B. shichuchen*; ^2^ *B. mosaicus*; ^3^
*B. cereus* sensu stricto; ^4^ *B. luti*; ^5^ *B. toyonensis*; ^6^ *B. mycoides*; ^7^ *B. pseudomycoides*; ^8^ *B. cytotoxicus*; ^9^ *B. paramycoides*. Dash denote no data available.

**Table 2 ijms-24-09636-t002:** Details of *B. shihchuchen* biovar anthracis pBS01 and other similar plasmids.

Strain/Plasmid/NCBI Accession Number	Genome Length (bp)	GC Content (%)	CDS	Isolation Source/ Disease	Isolation Country	Distance
*B. shihchuchen* biovar anthracis/pBS01/PRJNA948896	229,197	31.93	395	Softshell turtle/Septicemia	Taiwan	0
*Bacillus* sp. SYJ/unnamed1/CP036355	218,649	32.36	362	Softshell turtle/Septicemia	China	0.009604
*B. cereus* AH820/pAH820_272/CP001285	272,145	33.64	324	Periodontal pocket, humane/Periodontal disease	Norway	0.067763
*B. cereus* ATCC 10987/pBc10987/AE017195	208,369	33.44	248	Human/Food poisoning	Canada	0.068379
*Bacillus cereus* H3081.97/pH308197_258/CP001166	258,484	34.14	265	Human/Food poisoning		0.069006
*B. thuringiensis* c25/unnamed1/CP022346	326,530	32.24	365	Soil	South Korea	0.073002
*B. cereus* NC7401/pNCcld/AP007210	270,082	34.16	288	Human/Food poisoning		0.074799
*B. cereus* M3/pBCM301/CP016317	229,356	33.73	234	Fermented food	South Korea	0.077859
*B. cereus* 03BB108/pBFI_2/CP009636	238,933	31.9	235	Dust	USA	0.080322
*B. cereus* 03BB102/p03BB102_179/CP001406	179,680	32.17	214	Human/Pneumonia	USA	0.088783
*B. cereus* G9241/pBCX01/CP009592	190,860	32.63	224	Human	USA	0.092618
*B. anthracis* 2002013094/pXO1/CP009901	184,436	32.55	228	Soil	USA	0.093196
*B. cereus* biovar anthracis CI/pCI-XO1/CP001747	181,907	32.54	228	Human/Anthrax		0.093196
*B. anthracis* A1075/pXO1/CM003248.1	182,638	32.52	221	Human/Anthrax	Chile	0.093783
*B. anthracis* Brazilian Vaccinal/pXO1/CM007716.1	181,684	32.53	225	Vaccine strain	Brazil	0.093783
*B. anthracis* HYU01/pXO1/CP008847	181,894	32.54	227	Soil	South Korea	0.093783
*B. anthracis* BA1015/pXO1/CP009543	181,707	32.52	227	Bovine	USA	0.093783
*B. anthracis* BA1035/pXO1/CP009699	181,892	32.54	219	Human	South Africa	0.093783
*B. anthracis* RA3/pXO1/CP009696	181,920	32.54	227	Bovine	France	0.093783
*B. anthracis* Ames BACI008/unnamed1/CP009980	181,674	32.53	226	Beefmaster heifer/Anthrax	USA	0.093783
*B. anthracis* Ames A0462/unnamed1/CP010793	181,677	32.53	224	Cattle/Anthrax	USA	0.093783
*B. anthracis* Pollino/pXO1/CP010814	181,682	32.53	224	Cattle	Italy	0.093783
*B. anthracis* A1144/pXO1/CP010853	181,663	32.53	221	Human/Anthrax	Argentina	0.093783
*B. anthracis* Larissa/pXO1/CP012520	181,658	32.53	225	Ovis	Greece	0.093783
*B. anthracis* Tangail-1/pXO1/CP015777	181,677	32.53	228	Environmental	Bangladesh	0.093783
*B. anthracis* SPV842_15/pXO1/CP019589	181,684	32.53	225	Bovine aborted fetus/Abortion	Brazil	0.093783
*B. anthracis* FDAARGOS_341/unnamed1/CP022045	181,618	32.53	231	Healthy cattle/Anthrax Vaccine	USA	0.093783
*B. anthracis* 14RA5914/pXO1/CP023002	198,129	32.56	244	Bovine	Germany	0.093783
*B. anthracis* A2012/pXO1/AE011190	181,677	32.53	240	Humane/Anthrax		0.093783
*B. anthracis* Sterne/pXO1/CP009540	181,624	32.53	225		USA	0.093783
*B. anthracis* Ames Ancestor; A2084/pXO1/AE017336	181,677	32.53	240	Human/Anthrax		0.093783
*B. anthracis* CDC 684/pXO1/CP001216	181,773	32.53	218	Human/Anthrax		0.093783
*B. anthracis* A0248/pXO1/CP001599	181,677	32.53	227	Human/Anthrax	USA	0.093783
*B. anthracis* H9401/BAP1/CP002092	181,700	32.53	220	Human/Cutaneous anthrax	South Korea	0.093783
*B. anthracis* SVA11/pXO1/CP006743	181,793	32.54	226	Human/Anthrax	Swedish	0.093783
*B. anthracis* 8903-G/pXO1/CM002402.1	181,673	32.52	224	Soil	Georgia	0.093783
*B. anthracis* V770-NP-1R/pXO1/CP009597	181,710	32.52	216	Bovine	USA	0.093783
*B. anthracis* Turkey32/Turkey32/CP009316	181,760	32.53	229	Human/Cutaneous anthrax	Turkey	0.093783
*B. anthracis* Ohio ACB/Ohio ACB 1/CP009340	181,369	32.51	219	Pig	USA	0.094378
*B. thuringiensis* YWC2-8/pYWC2-8-1/CP013056	250,706	33.49	273	Soil	China	0.12214
*B. thuringiensis* serovar indiana HD521/pBTHD521/CP010111	253,580	33.08	258	Soil	USA	0.129604

**Table 3 ijms-24-09636-t003:** *B. shihchuchen* biovar anthracis QF108-045 multilocus sequence typing of different housekeeping genes.

Locus	Identity (%)	Coverage (%)	Alignment Length (bp)	Allele Length (bp)	Gaps	Allele
Glp *	99.7312	100	372	372	0	Glp_74 *
Gmk	100	100	504	504	0	Gmk_55
Ilv	100	100	393	393	0	Ilv_86
Pta *	99.7585	100	414	414	0	Pta_21 *
Pur	100	100	348	348	0	Pur_97
Pyc	100	100	363	363	0	Pyc_80
Tpi	100	100	435	435	0	Tpi_5

Notes: * denote alleles with less than 100% identity found; Glp *: novel allele, ST may indicate nearest ST; Pta *: novel allele, ST may indicate nearest ST.

**Table 4 ijms-24-09636-t004:** Antimicrobial susceptibility of isolate *Bacillus shihchuchen* biovar anthracis QF108-045.

Category	Antibiotic (Concentration)	Concentration (μg)	DIZ/mm	Standard DIZ			Susceptibility
				R	I	S	
Penicillin	Penicillin	10 unit	0	28		29	R
	Amoxicillin	25	0				R
	Ampicillin *	10	0	16		17	R
	Piperacillin	100	18	17		18	S
cephalosporin	Ceftifour	20	0				R
	Cephalexin	30	0				R
	Cephazolin	30	0	14	15~17	18	R
aminoglycosides	Neomycin	10	10				-
	Gentamycin	10	10	12	13~14	15	R
	Streptomycin	60	15				-
	Amikacin	30	14	14	15~16	17	R
	Tobramycin	10	11	12	13~14	15	R
Nitrofurans	Nitrofurantoin	30	12	14	15~16	17	R
Quinolone	Enrofloxacin	5	19				-
	Norfloxacin	10	17	12	13~16	17	S
	Ofloxacin	5	17	12	13~15	16	S
Glycopeptides	Vancomycin *	30	10	14		17	R
Tetracyclines	Tetracycline	30	10	14	15-18	19	R
	Oxytetracycline	30	18~28				-
	Doxytetracycline	30	28~30	12	13~15	16	S
Phenicols	Chloramphenicol	50	20	13	13~17	18	S
	Florfenicol	30	32				-
	Thiamphenicol	20	0				-
Macrolides	Erythromycin	15	28	13	14~22	23	S
	Clindamycin	2	16	14	15~20	21	I

Note: * Standard DIZ applied from *Enterococcus* spp. in CLSI. Dash represents susceptibility cannot be interpreted. Abbreviations: DIZ, Diameter of Inhibited Zone; R, Resistant; I, intermediate; S, Sensitive.

## Data Availability

The QF108-045 genome can be reached through NCBI BioProject accession number PRJNA948896.

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
