# Peer review of "Pathogenicity and Genomic Characterization of a Novel Genospecies, Bacillus shihchuchen, of the Bacillus cereus Group Isolated from Chinese Softshell Turtle (Pelodiscus sinensis)"

_ijms, 2023, doi:10.3390/ijms24119636_

Round 1

Reviewer 1 Report

Dear authors

The article on title “Pathogenicity and genomic characterization of a novel genospecies, Bacillus shihchuchen, of the Bacillus cereus group isolated from Chinese softshell turtle (Pelodiscus sinensis) by Li-Wu Cheng et al., lays out a very interesting connection between the emergence of  a pathogenic bacteria isolated from the Chinese softshell turtle (CST; Pelodiscus sinensis) with  high mortality rate and related with Bacillus cereus sensu lato group. According with whole-genome sequencing and pathogenicity analysis, the authors suggest a new genospecie that they named Bacillus shihchuchen. The manuscript is well performed and easy to follow. I think that this provides important results about the knowledge and futures perspectives for microbial detection of Bacillus shihchuchen and evaluation of antimicrobial test in Chinese softshell turtle. I recommend for acceptance with comments addressed below: 

1.-Abstract is adequately described.

2.- The introduction provide sufficient background and include relevant references. However, information about similar plasmids and virulence genes distributed among strains of the B. cereus Group is necessary.

3.- The methodology is adequately described, however, is necessary that in table 1 show differences in genome features, plasmids features, GC content with others related B. cereus s.l strains, as well as  is necessary a core and pangenome analysis with orthologous proteins differences between related strains. In addition, a deep genomic islands analysis is necessary between related strains as well as a comparative genomic of the plasmids. Analysis for PlcR regulon and motility genes could be better also to give a new specie. Probably, after these analyses the Bacillus shihchuchen specie name could be reconsidered.

4.-  Therefore, a major discussion is necessary with these analysis.

Author Response

Manuscript ID ijms-2387931

Responses to Reviewers

Dear reviewers,

Thank you for giving us the opportunity to submit a revised draft of the manuscript “"

Pathogenicity and genomic characterization of a novel genospecies, Bacillus shihchuchen, of the Bacillus cereus group isolated from Chinese softshell turtle (Pelodiscus sinensis)"” for publication in the international journal of molecular sciences (ijms-2387931). We appreciate the time and effort that reviewers dedicated to providing insightful comments on our manuscript. We have incorporated most of the suggestions made by the reviewers. Please see below, in question and answer format, for a point-by-point response to the reviewers’ comments and concerns. All page numbers refer to the revised manuscript.

Referee: 1

The article on title “Pathogenicity and genomic characterization of a novel genospecies, Bacillus shihchuchen, of the Bacillus cereus group isolated from Chinese softshell turtle (Pelodiscus sinensis) by Li-Wu Cheng et al., lays out a very interesting connection between the emergence of  a pathogenic bacteria isolated from the Chinese softshell turtle (CST; Pelodiscus sinensis) with  high mortality rate and related with Bacillus cereus sensu lato group. According with whole-genome sequencing and pathogenicity analysis, the authors suggest a new genospecie that they named Bacillus shihchuchen. The manuscript is well performed and easy to follow. I think that this provides important results about the knowledge and futures perspectives for microbial detection of Bacillus shihchuchen and evaluation of antimicrobial test in Chinese softshell turtle. I recommend for acceptance with comments addressed below:

Question 1: Abstract is adequately described.

Answer: We appreciate reviewer's comments.

Question 2: The introduction provide sufficient background and include relevant references. However, information about similar plasmids and virulence genes distributed among strains of the B. cereus Group is necessary.

Answer : We agree reviewer's comments.  We have added the information about similar plasmids and virulence genes distributed among strains of the B. cereus s.l in introduction part, L70~L77.

Question 3: The methodology is adequately described, however, is necessary that in table 1 show differences in genome features, plasmids features, GC content with others related B. cereus s.l strains, as well as is necessary a core and pangenome analysis with orthologous proteins differences between related strains. In addition, a deep genomic islands analysis is necessary between related strains as well as a comparative genomic of the plasmids. Analysis for PlcR regulon and motility genes could be better also to give a new specie. Probably, after these analyses the Bacillus shihchuchen specie name could be reconsidered.

Answer : We agree reviewer's comments.  We have added the genome features, plasmids features with others related B. cereus s.l strains in Table 1 and Table 2, respectively. GC content was added in Figure 2. However, core and pangenome analysis, deep genomic islands analysis, and comparative genomic of the plasmids will be our future work.

Question 4: Therefore, a major discussion is necessary with these analysis.

Answer : We agree reviewer's comments. We have added new discussion in L520~L524, L563~L569.

Reviewer 2 Report

The work by Chen et al., (Manuscript ID: ijms-2387931) entitled “Pathogenicity and genomic characterization of a novel genospecies, Bacillus shihchuchen, of the Bacillus cereus group isolated from Chinese softshell turtle (Pelodiscussinensis)”,is an interesting and full-bodied work concerning pathogenesis and genomic characterization of Bacillus cereus group strain in the Chinese softshell turtle. The performed analysis is accurate using valid methods and the manuscript is generally well written.Overall, although a multidisciplinary approach that combines molecular biology, comparative genomics, and functional analysis has been performed, genetic characterization of a new species of the Bacillus cereus group is doubtedlyto gain a comprehensive understanding of its genetic makeup. Thus, my main concern in this work is the authors’ statement about a distinct species. Even if the adaptation of the bacterium to the turtle could potentially lead to speciation, it must be clarified if the appropriate comparison with an already known strain of Bacillus thuringiensis(or other e.g. Bacillus wiedemannii, etc.)has been carried out. The phylogenetic analysis seems poor, as by using a combination of genetic markers and comparative genomics, authors could construct a phylogenetic tree that would show the evolutionary relationships between the potentially new and other known species in the Bacillus cereus group. This can help to determine whether the new species represents a distinct lineage or is closely related to known strains. Αdditionally, the QF108-045 genome could not be reached through NCBI BioProject accession number PRJNA948896 by the reviewer.Finally, many important references concerning the classification of Bacillus cereus group are missing and must be added prior to manuscript’s acceptance for publication in IJMS.

Author Response

Manuscript ID ijms-2387931

Responses to Reviewers

Dear reviewers,

Thank you for giving us the opportunity to submit a revised draft of the manuscript “"

Pathogenicity and genomic characterization of a novel genospecies, Bacillus shihchuchen, of the Bacillus cereus group isolated from Chinese softshell turtle (Pelodiscus sinensis)"” for publication in the international journal of molecular sciences (ijms-2387931). We appreciate the time and effort that reviewers dedicated to providing insightful comments on our manuscript. We have incorporated most of the suggestions made by the reviewers. Please see below, in question and answer format, for a point-by-point response to the reviewers’ comments and concerns. All page numbers refer to the revised manuscript.

Referee: 2

The work by Chen et al., (Manuscript ID: ijms-2387931) entitled “Pathogenicity and genomic characterization of a novel genospecies, Bacillus shihchuchen, of the Bacillus cereus group isolated from Chinese softshell turtle (Pelodiscus sinensis)”,is an interesting and full-bodied work concerning pathogenesis and genomic characterization of Bacillus cereus group strain in the Chinese softshell turtle. The performed analysis is accurate using valid methods and the manuscript is generally well written. Overall, although a multidisciplinary approach that combines molecular biology, comparative genomics, and functional analysis has been performed, genetic characterization of a new species of the Bacillus cereus group is doubtedly to gain a comprehensive understanding of its genetic makeup. Thus, my main concern in this work is the authors’ statement about a distinct species. Even if the adaptation of the bacterium to the turtle could potentially lead to speciation, it must be clarified if the appropriate comparison with an already known strain of Bacillus thuringiensis (or other e.g. Bacillus wiedemannii, etc.) has been carried out. The phylogenetic analysis seems poor, as by using a combination of genetic markers and comparative genomics, authors could construct a phylogenetic tree that would show the evolutionary relationships between the potentially new and other known species in the Bacillus cereus group. This can help to determine whether the new species represents a distinct lineage or is closely related to known strains. Αdditionally, the QF108-045 genome could not be reached through NCBI BioProject accession number PRJNA948896 by the reviewer. Finally, many important references concerning the classification of Bacillus cereus group are missing and must be added prior to manuscript’s acceptance for publication in IJMS.

Question 1: It must be clarified if the appropriate comparison with an already known strain of Bacillus thuringiensis (or other e.g. Bacillus wiedemannii, etc.) has been carried out.

Answer : We appreciate reviewer's comments. Due to Bcg exhibit a phylogenetic similarity while demonstrating significant ecological and phenotypic variations [1, 2], an accurate ontological framework for Bcg was built that adheres to widely accepted genomic and taxonomic definitions of bacterial genomospecies [3] while remaining insightful, straightforward, and comprehensible to those in public health by adding the biovar name (phenotype). Among which, Bacillus mosaicus genospecies included B. anthracis Ames (now B. mosaicus subsp. anthracis strain Ames), Bacillus mobilis (now B. mosaicus strain 0711P9-1), Bacillus pacificus (now B. mosaicus strain EB422), Bacillus paranthracis (now B. mosaicus strain MN5), Bacillus tropicus (now B. mosaicus strain N24), and Bacillus wiedmannii (now B. mosaicus strain FSL W8-0169) (L528~L532) [3]. In this study, we have compared Bacillus shihchuchen genomes with other Bcg reference strain, the result all showed lower than 95% ANI and 70% of dDDH. So we proposed Bacillus shihchuchen as a new genosepices.

Question 2: The phylogenetic analysis seems poor, as by using a combination of genetic markers and comparative genomics, authors could construct a phylogenetic tree that would show the evolutionary relationships between the potentially new and other known species in the Bacillus cereus group.

Answer : We appreciate reviewer's comments. We conduct phylogenetic analysis based on PATRIC phylogenetic tree building service, which was built on 290 genes in each reference strain. The detail of statistics’ information of phylogenetic tree were listed in Table S8 and Table S9, and detail of alignment methodology were in L717~L 728

Question 3: The QF108-045 genome could not be reached through NCBI BioProject accession number PRJNA948896 by the reviewer.

Answer : We appreciate reviewer's comments. We choose to release data when research article has been accepted in SCI journal. Once paper been published, NCBI will automatically publish BioProject PRJNA948896 related information.

Question 4: Finally, many important references concerning the classification of Bacillus cereus group are missing and must be added prior to manuscript’s acceptance for publication in IJMS.

Answer : We appreciate reviewer's comments. Since new Bacillus cereus group taxonomic nomenclature was recently published, there is less reference in this field [3]. However, there are some new genospecie been published based on ANI and dDDH method, which have been included in the MS (L527~L537) [4-5].

Reference:

  1. Okinaka, R. T.; Keim, P., The Phylogeny of Bacillus cereus sensu lato. Microbiology spectrum 2016, 4, (1).
  2. Fayad, N.; Kallassy Awad, M.; Mahillon, J., Diversity of Bacillus cereus sensu lato mobilome. BMC Genomics 2019, 20, (1), 436.
  3. Carroll Laura, M.; Wiedmann, M.; Kovac, J., Proposal of a Taxonomic Nomenclature for the Bacillus cereus Group Which Reconciles Genomic Definitions of Bacterial Species with Clinical and Industrial Phenotypes. mBio 2020, 11, (1), e00034-20.
  4. Jiménez, G.; Urdiain, M.; Cifuentes, A.; López-López, A.; Blanch, A. R.; Tamames, J.; Kämpfer, P.; Kolstø, A.-B.; Ramón, D.; Martínez, J. F.; Codoñer, F. M.; Rosselló-Móra, R., Description of Bacillus toyonensis sp. nov., a novel species of the Bacillus cereus group, and pairwise genome comparisons of the species of the group by means of ANI calculations. Systematic and Applied Microbiology 2013, 36, (6), 383-391.
  5. Guinebretière, M. H.; Auger, S.; Galleron, N.; Contzen, M.; De Sarrau, B.; De Buyser, M. L.; Lamberet, G.; Fagerlund, A.; Granum, P. E.; Lereclus, D.; De Vos, P.; Nguyen-The, C.; Sorokin, A., Bacillus cytotoxicus sp. nov. is a novel thermotolerant species of the Bacillus cereus Group occasionally associated with food poisoning. Int J Syst Evol Microbiol 2013, 63, (Pt 1), 31-40.
